# Direct control of CAR T cells through small molecule-regulated antibodies

Spencer Park[1,2,8], Edward Pascua[1,8], Kevin C. Lindquist [1,8], Christopher Kimberlin [1,3], Xiaodi Deng[1,4], Yvonne S. L. Mak[1,5], Zea Melton[1,5], Theodore O. Johnson[1], Regina Lin[1,5], Bijan Boldajipour[1,2], Robert T. Abraham[1,6], Jaume Pons[1,7], Barbra Johnson Sasu [1,5], Thomas J. Van Blarcom [1,5✉] & Javier Chaparro-Riggers [1✉]

Antibody-based therapeutics have experienced a rapid growth in recent years and are now utilized in various modalities spanning from conventional antibodies, antibody-drug conjugates, bispecific antibodies to chimeric antigen receptor (CAR) T cells. Many next generation antibody therapeutics achieve enhanced potency but often increase the risk of adverse events. Antibody scaffolds capable of exhibiting inducible affinities could reduce the risk of adverse events by enabling a transient suspension of antibody activity. To demonstrate this, we develop conditionally activated, single-module CARs, in which tumor antigen recognition is directly modulated by an FDA-approved small molecule drug. The resulting CAR T cells demonstrate specific cytotoxicity of tumor cells comparable to that of traditional CARs, but the cytotoxicity is reversibly attenuated by the addition of the small molecule. The exogenous control of conditional CAR T cell activity allows continual modulation of therapeutic activity to improve the safety profile of CAR T cells across all disease indications.

[1] Pfizer, La Jolla, CA, USA. [2]Present address: Lyell Immunopharma, South San Francisco, CA, USA. [3]Present address: Asher Bio, South San Francisco, CA, USA. [4]Present address: Dren Bio, San Carlos, CA, USA. [5]Present address: Allogene Therapeutics, South San Francisco, CA, USA. [6]Present address: Vividion Therapeutics, San Diego, CA, USA. [7]Present address: ALX Oncology, Burlingame, CA, USA. [8]These authors contributed equally: Spencer Park, Edward Pascua, Kevin C. Lindquist. ✉email: tom.vanblarcom@allogene.com; javier.chaparro-riggers@pfizer.com

Therapeutic monoclonal antibodies have become a prevalent treatment modality for various diseases over the past 25 years, with 48 new monoclonal antibodies being approved since 2008[1]. Recently, many novel antibody-based modalities have been developed to enhance potency while taking advantage of antibody specificity. These modalities range from antibody–drug conjugates and bispecific antibodies to CAR T cells. Amplified potency, however, has often been associated with an increased risk of adverse events[2]. Therefore, technologies that enable rapid and reversible modulation of antibody activity in vivo are of great interest.

Treatment of hematologic malignancies has been revolutionized by the genetic manipulation of T cells to express CARs, as manifested by remarkable clinical efficacy[3–7]. However, when treating patients with high tumor burden, simultaneous target engagement by the infused CAR T cells can trigger extensive expansion and activation, resulting in adverse events that can lead to significant morbidity, therefore hindering the general use of this curative treatment option. Cytokine release syndrome (CRS), the most common toxicity observed for CAR T cell therapy, is a cascade of immunological events initiated by the synchronous release of cytokines from overactivated T cells, such as IFN-γ and IL-2, which in turn activate neighboring myeloid cells and macrophages to release additional inflammatory cytokines, for instance, IL-6[8,9]. In addition to CRS, neurotoxicity and tumor lysis syndrome are two other prominent CAR T cell therapy-associated side effects, where elevated levels of cytokines in the brain and massive release of intracellular tumor contents lead to serious metabolic disorders[10–13]. Current clinical management of CRS relies on corticosteroids and/or neutralization of IL-6 signaling with an anti-IL-6R antibody to reduce the supraphysiological inflammatory response. While these methods are generally effective, they often require hours to take effect and in some cases symptoms do not completely resolve, warranting further medical intervention[14,15].

Targeting the root cause of such toxicities through the direct modulation of CAR T cell activity is a highly desirable approach and a variety of innovative technologies have been developed to achieve this[16]. Many of these technologies have focused on protein-based control elements over those that operate at the genetic level due to their superior response times[17,18], a critical requirement for treating adverse events. One popular approach uses "suicide-switches" to eliminate the adoptively transferred cells[19–21]. While this approach reduces toxicity, it also prematurely terminates antitumor efficacy. Moreover, the heterogeneous expression of the switch results in incomplete CAR T cell elimination[20]. A more attractive strategy is to design CAR T cells with a secondary de-activation module that rapidly responds to an exogenous input[22,23], thus allowing physicians to control the activity of infused cells according to the patients' response without the risk of permanently curtailing the therapeutic effect. However, such multi-module approaches that require the expression of additional proteins are more difficult to implement due to the limited maximum payload capacity of retrovirus and lentivirus[24,25].

Incorporation of a small molecule-based control element into the antigen receptor of the CAR would enable physicians to rapidly and transiently control CAR T cell activity without increasing the viral payload size, making the application of such elements more universal. This approach calls for a bifunctional antibody scaffold capable of binding to different tumor-associated antigens (TAAs) and a small molecule. This is a paradox since antibodies binding to TAAs tend to utilize large, planar-binding sites[26] while antibodies binding to small molecules tend to utilize small cavities that are deeply buried in their variable heavy-chain domain ($V_H$):variable light-chain domain ($V_L$) interface, which is inaccessible to large proteins[26].

In this work, we combine these two disparate binding mechanisms by using a camelid antibody ($V_{HH}$) with unique properties[27] as a scaffold to develop conditionally activated, single-module CARs, in which TAA recognition is directly modulated by the FDA-approved small molecule drug methotrexate (MTX).

## Results

**Generation of conditionally active antibodies.** $V_{HH}$ antibodies only possess a single-variable domain containing three complementarity determining regions (CDRs; CDRH1, CDRH2, and CDRH3) that drive their high degree of antigen specificity compared to traditional antibodies that are comprised of two variable domains ($V_H$ and $V_L$), containing six CDRs (CDRH1, CDRH2, CDRH3, CDRL1, CDRL2, and CDRL3). The crystal structures of the aforementioned anti-MTX $V_{HH}$ with and without bound MTX suggest CDRH1, CDRH2, and framework 3 (FW3) of the $V_{HH}$ are important for MTX recognition, and CDRH1 and FW3 undergo a conformational change upon binding MTX[27]. This unique binding mode leaves the possibility of engineering CDRH3, the typical key determinant of antibody specificity, to bind TAAs while taking advantage of the MTX-induced conformational change to disrupt TAA binding. Therefore, the $V_{HH}$ could be used as a basis of a conditionally active antibody scaffold ($V_{HH}$-MTX) for the engineering of antibodies that bind a diverse set of TAAs, whose activity is regulated by the presence or absence of MTX.

Since antibodies typically require multiple CDRs to achieve high specificity and affinity[28], we converted the $V_{HH}$-MTX scaffold into a traditional antibody scaffold by combining it with a variable light-chain domain to provide three additional light-chain CDRs to be engineered. To achieve this, CDRH1, CDRH2, and FW3 regions of the $V_{HH}$ were grafted onto the highly homologous $V_H$ from the human antibody M2J1[29] (PDB entry: 3F12) to maintain MTX binding[27], and fused to a flexible linker and the $V_L$ of M2J1 (Supplementary Fig. 1a). The resulting single chain antibody fragment (scFv; M2J1-MTX) successfully bound MTX albeit with reduced affinity compared to the parental $V_{HH}$-MTX (Fig. 1a). While MTX binding was heavily dependent upon most of the residues in CDRH2 and the CAA motif at the N-terminus of CDRH3 (Supplementary Fig. 2), it tolerated substitutions elsewhere including the entire $V_L$ (Fig. 1b and Supplementary Fig. 1b) and residues 95 through 102 in CDRH3 thereby providing four CDRs to engineer TAA specificity without abrogating MTX binding.

To evolve the scFv to possess specificity toward both MTX and a TAA, we designed a synthetic human antibody library based on M2J1-MTX scFv by fixing CDRH1 and CDRH2 to maintain MTX binding. CDRH3 was randomized and combined with three different light chains, each with fully randomized CDRs in a manner similar to that previously described[30,31] (Fig. 1c). The scFv library was displayed on phage, and through several rounds of biopanning toward the TAA (CD33 or EGFR), conditional scFvs that bound the TAA in the absence of MTX and displayed reduced binding in the presence of MTX were identified (Supplementary Figs. 3, 4). A panel of 14 anti-CD33 scFvs was expressed as scFv-Fc and further characterized. The percent inhibition at 10 μM MTX concentration was determined for each scFv and values ranged from 23.3% to 99.7% (Supplementary Fig. 3a, b). Inhibition of TAA binding by MTX was dose-dependent (Supplementary Fig. 3c). The scFv binding affinity for CD33 was determined, with equilibrium dissociation constant ($K_D$) values ranging from 0.265 to 84.4 nM (Supplementary Fig. 3d, e). The affinities for MTX binding were also determined and $K_D$ values ranged from 81.9 nM to 5.06 μM (Supplementary Fig. 3f, g).

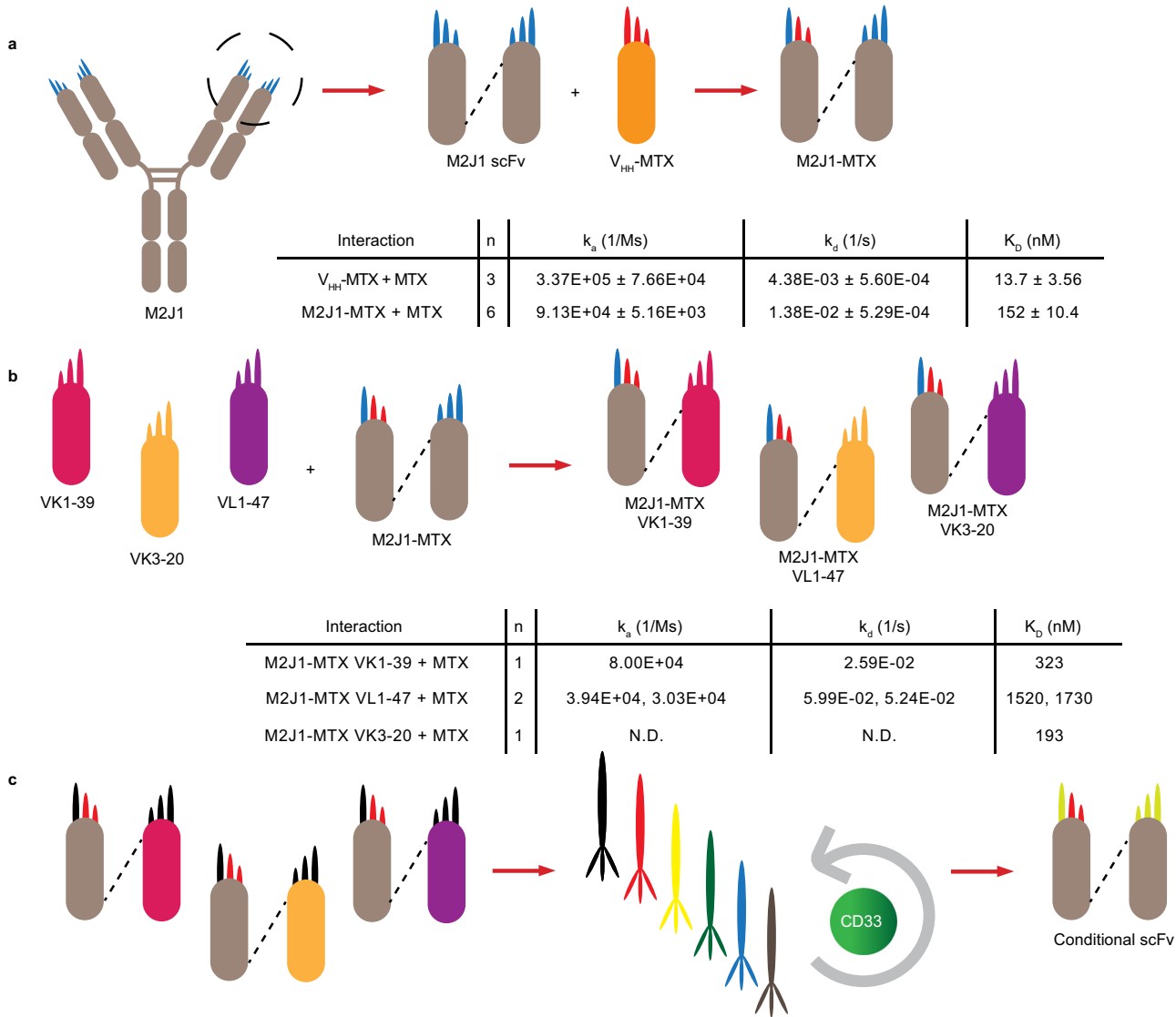

**Fig. 1 Design of conditional scFv and screening. a** CDRH1 (in red), CDRH2 (in red) and FW3 from $V_{HH}$-MTX were grafted onto the $V_H$ of scFv derived from M2J1 antibody (M2J1 scFv) to form a humanized M2J1-MTX scaffold (M2J1 CDRs in blue). **b** $V_H$ of M2J1-MTX can be paired with different germline $V_L$'s and maintain binding to MTX. N.D. is not determined. **c** A conditional scFv library was generated by fixing CDRH1 and CDRH2 and diversifying the amino acid sequences of CDRH3 and all three light-chain CDRs. This library was screened using phage display to isolate clones that bound CD33 in an MTX-dependent manner.

**Structure of conditional scFv resembles parental $V_{HH}$.** In order to determine if the molecular mechanism of MTX binding in the $V_{HH}$ was maintained after pairing it with an antibody light-chain variable domain, we determined the structures of a conditional scFv in the absence and presence of MTX using anti-CD33 scFv P02_D09, a highly MTX-sensitive scFv (Supplementary Fig. 3). Comparison of the scFv with the original $V_{HH}$-MTX scaffold, upon which the conditional scFv is based, revealed that the MTX binding pocket is largely maintained, conserving the key interactions of MTX with CDRH1 and FW3 without being perturbed by the diversification of CDRH3 and addition of the light chain (Fig. 2a, b). While not directly engaged in the binding of MTX, CDRH2 makes contributions to the stability of the MTX binding pocket formed by the framework and CDRH1. In particular, the carbonyl groups of S52 and Y53 orient the side chains of FW3 residues R72 and N74 to form hydrogen bonds with the pterin moiety of MTX (Supplementary Fig. 5). In addition, the $V_H:V_L$ interface seen in both the apo and MTX-bound scFv structures are similar to the $V_H:V_L$ interface in the structure of parental M2J1 antibody. Mutation of select amino acids of the $V_{HH}$ framework back to the parental M2J1 sequence successfully restore the $V_H:V_L$ interface with sidechain positions essentially superimposable between structures (Fig. 2c and Supplementary Fig. 1a). As expected, binding of MTX to the scFv resulted in a conformational change of the paratope: in the absence of MTX, W32 is oriented outwards and faces the light chain, while upon MTX binding it rotates inwards to make contact with MTX within the IgV core creating additional space for significant movement of CDRH3 as demonstrated by a 7.1 Å (C-α to C-α) shift of D102 (Fig. 2d).

**Conditional CAR T cell activity can be modulated by MTX.** Next, we sought to implement the MTX-conditional antibodies in a CAR with the goal of creating MTX-conditional CAR T cells. The conditional scFvs exhibiting the largest affinity reduction toward CD33 in the presence of MTX were selected and cloned into a second-generation CAR lentiviral vector[32] (Fig. 3a).

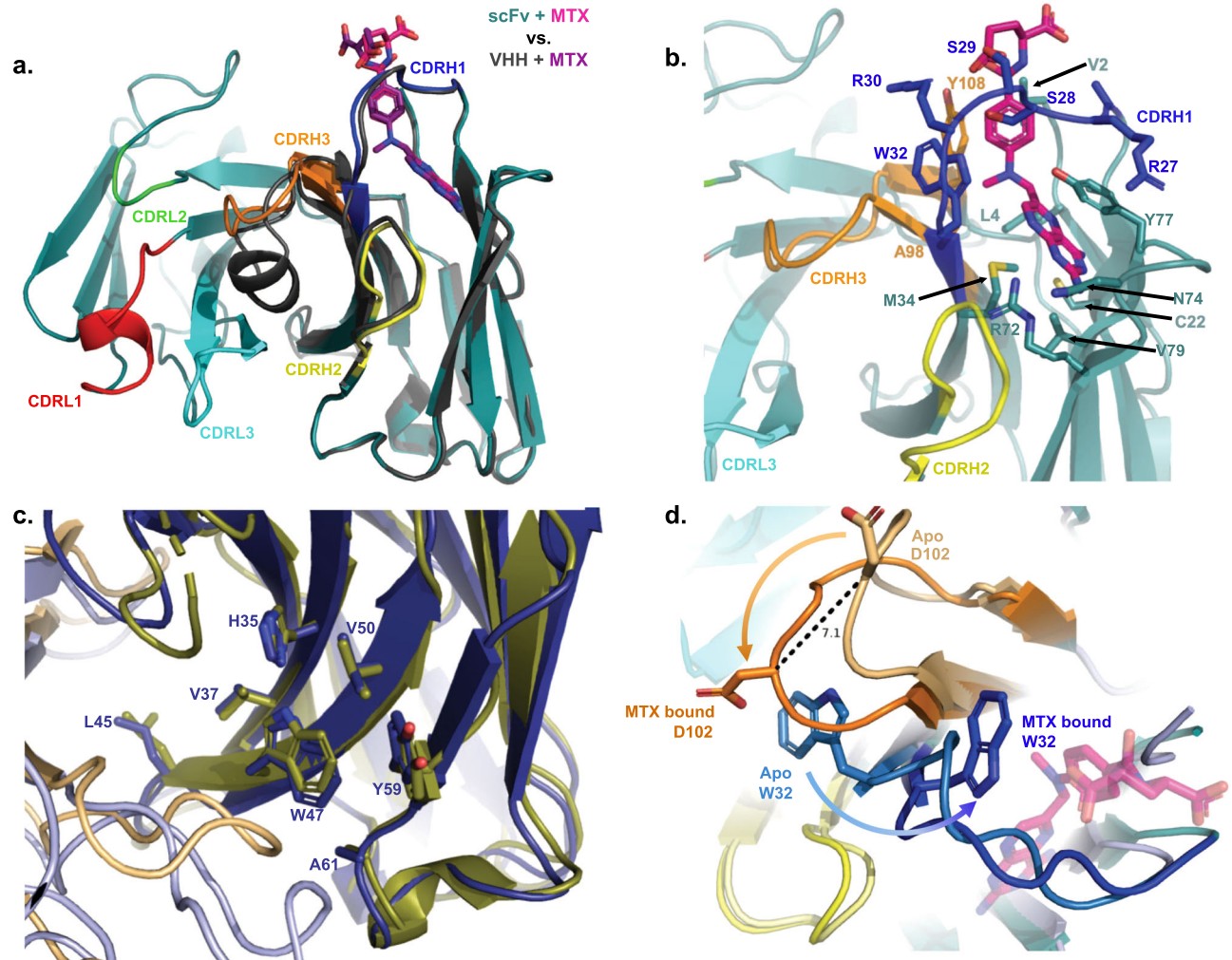

**Fig. 2 Structures of the conditional scFv PO2_DO9 were determined with or without MTX. a** Comparison with the previous $V_{HH}$ structure bound to MTX (PDB code: 3QXV) demonstrates the presence of the light chain does not significantly perturb the MTX binding site. Structures are shown in cartoon rendering with scFv framework in deep teal, CDRHs 1, 2, and 3 shown in blue, yellow, and orange, respectively, CDRLs 1, 2, and 3 shown in red, green, and cyan, respectively, and MTX shown as magenta sticks. $V_{HH}$ + MTX is shown in a black cartoon with MTX as purple sticks. **b** A more detailed view of MTX bound to the conditional scFv shows MTX sitting underneath CDRH1 while also interacting with FW3. Residues contacting MTX are noted in the figure. **c** Alignment of the apo scFv structure (deep blue, HC and light blue, LC) with parental antibody M2J1 (deep olive, HC and light orange, LC) highlighting the interface between $V_H$ and $V_L$. Side chains of mutations that revert the original $V_{HH}$ sequence back to the M2J1 sequence to stabilize the interface are shown as sticks. **d** Zoomed in view of the transition between apo and MTX-bound scFv PO2_DO9 highlighting the inward flip of W32 in CDRH1 upon MTX binding and the corresponding movement of D102 in CDRH3. Structures are shown as cartoon and colored as in (**a**) with noted side chains and MTX shown as sticks.

Lentivirus containing these conditional CAR (condCAR) clones, as well as a conventional CAR (CAR) targeting human CD33, were used to transduce human primary T cells (Supplementary Fig. 6). The resulting condCAR T cells demonstrated target-specific cytotoxicity comparable to that of conventional CAR T cells (Fig. 3b and Supplementary Fig. 7).

In oncology settings, MTX is being used to competitively inhibit dihydrofolate reductase (DHFR) activity in folate synthesis, leading to impaired DNA synthesis and cellular replication[33]. However, here in the conditional CAR platform, MTX is intended as a small molecule switch acting extracellularly, and not as an intracellular cytotoxic drug. As a result, its inherent toxicity complicates evaluating an MTX-controlled off-switch in cellular assays. We partially addressed this complication by adding leucovorin (LV), an FDA-approved drug for combating high-dose MTX toxicity. The addition of LV increased the viable population of activated CAR T cells (70% vs 12% viable, 10 µM MTX ± LV) and the CD33-expressing tumor cell line MV4-11 (65% vs 11%

viable, 10 µM MTX ± LV) (Supplementary Fig. 8). To further reduce MTX-mediated cellular toxicity, T cells and MV4-11 cells were engineered to express a DHFR mutant (MV4-11mut) that is 4000-fold less sensitive to MTX[34], allowing normal cellular functions in the presence of MTX (Supplementary Fig. 8). While the addition of the DHFR mutant in tumor cells resulted in a pronounced reduction of toxicity (106% viable MV4-11mut cells vs. 65% parental, 10 µM MTX + LV), it was only marginally beneficial in CAR T cells (82% viable DHFR mutant vs. 70%, 10 µM MTX + LV). Consequently, we assessed the effect of MTX on CAR T cell-mediated cytotoxicity in the presence of LV using tumor cells containing the DHFR mutant.

Next, we determined the effect of MTX on condCARTs cytotoxicity, activation, and subsequent cytokine secretion. First, the addition of MTX to all the condCAR T cells decreased target lysis (Fig. 3b and Supplementary Fig. 9), confirming the conditional activity. In particular, clone PO2_DO9-CAR, which was highly efficient in target cell lysis in the absence of MTX,

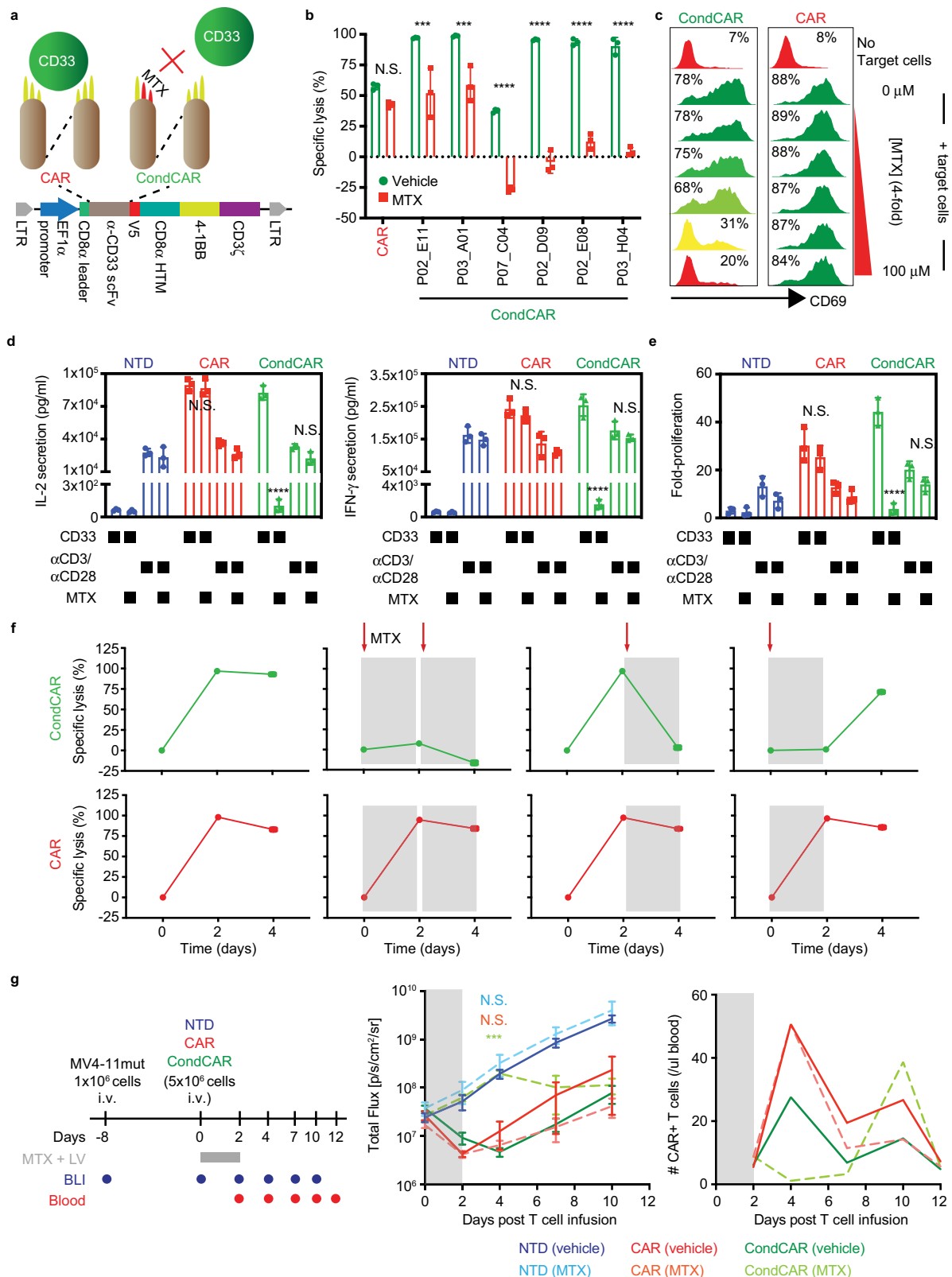

displayed significantly attenuated activity in the presence of MTX at concentrations of 12.5 μM and above (Supplementary Fig. 9). Similar MTX-induced disruption of conditional anti-EGFR CAR T cell-mediated cytotoxicity was also observed with conditional anti-EGFR CAR T cells (Supplementary Fig. 10). Conventional CAR T cells cytotoxicity was unaffected by MTX (Fig. 3b and Supplementary Fig. 9), confirming the circumvention of MTX-

associated suppressive effects in T cells. Second, we determined the impact of MTX on condCAR T cell activation and subsequent cytokine secretion and T cell proliferation, both of which are closely correlated with eliciting CRS. In the presence of target cells expressing CD33, condCAR T cells and conventional CAR T cells showed substantial activation after 24 h (78 and 88% CD69+ compared to 7 and 8% in the absence of target cells, respectively).

**Fig. 3 Conditional CAR T cells are as effective as conventional CAR T cells and can be modulated by MTX. a** Schematic of CAR constructs. **b** Antigen-specific cytotoxicity was determined after a 48 h co-incubation of MV4-11mut cells with conventional CAR or condCAR T cells ± MTX ($n = 3$ biologically independent samples, mean ± SD). Effector:target (E:T) = 1:1. **c** MTX- and CD33-dependent T cell activation was quantified by measuring the surface expression of CD69. Partial (yellow) or complete (red) inhibition of CD69 expression is achieved at or above 10 μM MTX, respectively. E:T = 1:2. **d** Antigen-specific secretion of the IL-2 and IFN-γ was quantified after a 24-h co-culture with either MV4-11mut cells or anti-CD3/anti-CD28-coated beads ± MTX ($n = 3$ independent experiments, mean ± SD). E:T = 1:2. **e** Antigen-specific proliferation was analyzed after 7 days of co-culture with either MV4-11mut (E:T = 1:5) or anti-CD3/anti-CD28-coated beads ± MTX ($n = 3$ biologically independent samples, mean ± SD). **f** Antigen-specific cytotoxicity of MV4-11mut cells by condCAR T cells can be modulated in a reversible manner. Kinetic analysis of cytotoxicity ± MTX was quantified every 2 days. The presence of MTX is shown by regions in gray ($n = 3$ technical triplicates, mean ± SD). E:T = 1:1. **g** Left panel: workflow used to assess the extent of condCAR T cell control by MTX in vivo. Two-day continuous infusion of MTX is indicated by the gray bar, and bioluminescence (BLI) analysis time points are indicated by blue dots. A parallel study was utilized for blood collection for CAR T cell enumeration with time points indicated by red dots. Middle panel: BLI analysis of MV4-11mut tumor model infused with condCAR T cells, conventional CAR T cells, or NTD cells ± 250 mg/day MTX (gray region) ($n = 5$ animals, mean ± SEM). Right panel: Circulating CAR T cell enumeration. One mouse was sacrificed at each time point for blood analysis by flow cytometry. *P* values in **b**, **d**, and **e** were calculated by paired two-tailed *t* test. *P* values in **g** were calculated by two-way ANOVA with Bonferroni posttest. N.S. non-significant ($P > 0.01$), \*\*\*$P < 0.001$, \*\*\*\*$P < 0.0001$.

However, MTX suppressed condCAR T cell activation in a dose-dependent manner with 100 μM MTX resulting in 20% activation of condCAR T and 84% activation of conventional CAR (Fig. 3c). Similarly, the high levels of secreted IL-2 and IFN-γ, two prominent serum markers for CRS, by the condCAR T cells (82.5 ng/ml IL-2, 253.8 ng/ml IFN-γ) were comparable to those of the conventional CAR T cells (89.7 ng/ml IL-2, 242.6 ng/ml IFN-γ) in the absence of MTX, but were dramatically reduced in the presence of 100 μM MTX (0.1 ng/ml IL-2, 1.6 ng/ml IFN-γ) and to levels comparable to non-transduced T cells (NTD; 0.1 ng/ml IL-2, 0.6 ng/ml IFN-γ; Fig. 3d). Target-specific proliferation, which is critical for CAR T cell-driven therapy as well as toxicities, was substantially lower only for the condCAR T cells in the presence of 100 μM MTX (fourfold with MTX vs. 44-fold without MTX) and again comparable to NTD (twofold with MTX vs. threefold without MTX) (Fig. 3e). Further investigation confirmed that the inhibitory effects were caused by obstruction of condCAR engagement, as the cytokine profile and the proliferative capacity were unaffected by MTX when condCAR T cells were stimulated via their TCRs (Fig. 3d, e).

Ideally, CAR T cell inhibition is reversible and can be accomplished without eliminating the engineered cells, such that therapy can resume following resolution of adverse events. Hence, we designed studies to evaluate the feasibility of a reversible "OFF-switch" mechanism. First, CAR T cells were incubated with the target cells in the absence of MTX, resulting in uninhibited cytotoxicity (Fig. 3f). At this point, additional target cells were added in fresh media with or without MTX supplementation. While target cell lysis continued in the absence of MTX, it was completely inhibited for condCAR T cells in the presence of MTX. Alternatively, CAR T cells can be gradually activated to minimize CRS, or other side effects related to synchronous engagement of high antigen density target cells, such as tumor lysis syndrome. To model this "ON-switch" approach, CAR T cells were initially co-cultured with MV4-11mut cells in the presence of MTX, which prevented condCAR T cells from engaging the target cells, but these MTX-treated CAR T cells regained the ability to lyse tumor cells when fresh media lacking MTX was introduced (Fig. 3f). Target lysis by conventional CAR T cells was not affected by the presence of MTX in both scenarios.

**Conditional CAR T cell antitumor effect can be modulated by MTX.** These encouraging data prompted us to design the following study to understand the in vivo activity of condCAR T cells in a switch-dependent manner (Fig. 3g). Mice with established MV4-11mut tumor burden received conventional CAR T cells, condCAR T cells, or NTD. In addition, a combination of MTX (250 mg/day) and LV was simultaneously

administered via osmotic pumps for the first two days (Day 0 to Day 2) in order to temporarily block recognition of tumor cells by condCAR T cells. Tumor burden was cleared to a similar extent in mice administered with condCAR T cells in the absence of MTX as those administered with conventional CAR T cells in the presence or absence of MTX, confirming that the applied MTX dose has no adverse effect on T cell activity in this context (Fig. 3g and Supplementary Fig. 11). However, in mice administered with condCAR T cells and MTX, tumor burden initially increased at a rate comparable to those treated with NTD for the two-day duration of MTX administration, demonstrating effective prevention of tumor engagement. Following removal of MTX, the tumor began to regress. On day 10, tumor burden for mice treated with the condCAR T cells (in the absence of MTX) was similar to that in mice treated with conventional CAR T cells. Since recent clinical data suggest a close correlation between CAR T cell expansion and induction of CRS[35], the circulating CAR T cells were enumerated over the course of this study (Fig. 3g). MTX treatment hindered the expansion of condCAR T cells and an increase in circulating T cell number was detected only after the removal of MTX. The delayed suppression of tumor growth immediately following the withdrawal of MTX was likely due to the time required for CAR T cells to sufficiently expand in number (Fig. 3g). Overall, this study demonstrates that the condCAR T cell design can achieve substantial in vivo switchable activity and may help avert CRS triggered by excessive cell activation and proliferation.

## Discussion

In summary, we generated conditional antibodies whose binding affinity to a target antigen is modulated by a small molecule. In this specific case, we selected MTX, an FDA-approved drug, and CD33, a well-studied and validated antigen for AML therapies, to highlight the application of the conditional antibody technology to CAR T cells. Implementing the single-module conditional CAR design allowed robust control over CAR T cell activity in cell-based assays as well as in mouse models, without impacting the size of the lentiviral cassette, which is a critical concern in cellular therapy manufacturing. Furthermore, we validated that the technology is applicable to other target antigens through discovering conditional antibodies against EGFR (Supplementary Fig. 4, Supplementary Fig. 10). Collectively, our results provide a general framework that can be broadly applied to the development of safer antibody-based therapeutics.

CRS is the most common toxicity observed for CAR T cell therapy and intervention at its onset typically shows resolution within a few hours to 2 days[36]. This timing aligns very well with established short-term (2–36 h) continuous infusion protocols for

MTX that can maintain MTX serum concentrations over the range necessary to transiently inhibit the activity of the condCAR T cells[37,38] until CRS subsides. Further, a prolonged or irreversible intervention is not desirable as the therapeutic activity of the condCAR T cells would be diminished during this time. To mimic this situation, we used an in vivo tumor model to demonstrate that dosing of MTX with leucovorin can transiently halt condCAR T cell activity over a 2-day duration, after which MTX can be removed to restore the antitumor effect.

While the use of MTX to transiently downmodulate a T cell-based therapies for cancer treatment is particularly appropriate since high-dose MTX chemotherapy followed by leucovorin rescue is a standard therapeutic regimen for a variety of adult and pediatric cancers, additional approaches could be undertaken to further address concerns regarding MTX-related toxicities and to allow for longer and/or higher dosing. For example, increasing the affinity of the conditional scFv toward MTX, while maintaining affinity toward the TAA, would increase the sensitivity of the condCARs and enable control of CAR T cells at lower MTX concentrations. Alternatively, the chemical composition of MTX can be modified to obtain analogs that maintain binding to the extracellular scFv and reduce toxicity by lowering the affinity of MTX to DHFR or by reducing cellular uptake. These structural analogs of MTX would circumvent the need for leucovorin supplementation and allow longer dosing at higher concentrations if needed. We have initiated an effort on the latter approach and successfully designed MTX structural analogs that retain the affinity toward the conditional scFv, but exhibit much reduced toxicity (Supplementary Fig. 12 and Supplementary Table 1). Collectively, we believe these efforts will aid the bench-to-bedside translation of this technology.

It was recently demonstrated in a preclinical setting that the FDA-approved small molecule dasatinib, which inhibits T cell receptor signaling, can be used to quickly and transiently modulate CAR T cell activity without eliminating them[39,40]. The use of dasatinib is similar to our approach being that the small molecule has antitumor activity, does not negatively impact CAR T cell manufacturing, and rapidly modulates CAR T cell activity in a reversible manner. However, unlike the approaches described above to specifically reduce MTX-associated toxicities without negatively impacting its desired "switch" function, reducing those associated with dasatinib without simultaneously diminishing its CAR T cell-suppressive effect will be challenging since it is a promiscuous kinase inhibitor that acts by inhibiting multiple fundamental cell signaling pathways[41], including those used by CAR T cells.

While our work focused on the reduction of toxicity for CAR-based therapies, other antibody-based drug modalities have also been associated with significant toxicities that could be reduced through the use of conditional antibodies. For example, CRS has been associated with T cell-redirecting bispecific antibodies[42,43], and tumor lysis syndrome has been associated with antibody–drug conjugates[44,45], both of which are especially pronounced in patients with high tumor burden. Autoimmune adverse events have been associated with monoclonal antibody-based immune checkpoint inhibitors that target PD-1 and CTLA-4[46]. For these therapeutic modalities, the difference in in vivo clearance between the antibody (slower clearance) and MTX (faster clearance) can be exploited to allow for rapid modulation of the antibody activity as needed to transiently reduce toxicity. Additional preclinical studies are warranted to explore dosing strategies and to justify the suitability of these approaches.

We present here, for the first time to our knowledge, conditionally active antibodies, whose binding affinity can be controlled by the addition of a small molecule. We demonstrated how these antibodies can be used to generate conditional CAR T cells with target-specific cytotoxicity whose activity can be reversibly attenuated by the addition of a small molecule to potentially mitigate therapy-associated toxicities. In addition to CAR T cells, these unique scaffolds could be applied to a broad range of antibody-based therapeutics for the development of safer therapeutics including, bispecific antibodies[43], antibody–drug conjugates[44,45], and antibody-based immune-modulatory agents[46], which are often hindered by a variety of associated adverse effects. Endowing exogenous control over antibody-based therapeutics would dramatically enhance their therapeutic index, thereby increasing their therapeutic value in tackling difficult-to-treat diseases.

## Methods

**Cell lines, media, and reagents**. HEK293T (ATCC) and MV4-11 (ATCC) were cultured in media per the supplier's recommendation. Expi293 (Thermo Fisher Scientific) was cultured in media per the supplier's recommendation. U87-EGFR is a kind gift from Cellectis SA (Paris, France). U87-EGFR was derived from the parental cell line, U87MG (ATCC) by first knocking out endogenous EGFR using Transcription Activator-Like Effector Nucleases (TALEN), and then stably over-expressing full-length human EGFR via lentiviral transduction. X-VIVO[TM] 15 was obtained from Lonza, rhIL-2 from Miltenyi Biotech, human serum AB from Seralab, human T-activator CD3/CD28 from Thermo Fisher Scientific, MACS® LD-column from Miltenyi Biotech, Hyclone FBS from GE Healthcare Life Sciences, Ficoll-Paque PLUS from GE Healthcare Life Sciences and Pan T cell Isolation Kit, human from Miltenyi Biotech. Methotrexate disodium salt and leucovorin calcium were obtained from Alfa Aesar and Sigma, respectively.

**Protein expression and purification**. The extracellular domains of human CD33 (positions 18–256) and human EGFR (positions 25–642) were cloned with a C-terminal TEV protease cleavage site, 8× polyhistidine, and AviTag[TM] (Avidity, LLC). For the scFvs, the heavy and light-chain variable domains were paired using a $(GGGGS)_4$ linker and fused to a C-terminal 8× polyhistidine tag. All constructs were expressed using the Expi293 system (Thermo Fisher Scientific) and purified by Ni Sepharose Excel column (GE Healthcare Life Sciences) followed by size-exclusion chromatography over a Superdex 75 or Superdex 200 column (GE Healthcare Life Sciences) equilibrated in 1× PBS (Corning). Site-specific biotinylation of CD33 through the AviTag[TM] was performed according to the manufacturer's protocols. scFv-Fc constructs reformatted from scFv libraries were purified by Protein A affinity chromatography (GE Healthcare Life Sciences).

**Conditional scFv library design and generation**. An existing Pfizer semi-synthetic human scFv antibody library[30] generated with Slonomics (Morphosys) was re-engineered to include an MTX-binding pocket. This was accomplished by grafting FW1, CDRH1, FW2 position 33, CDRH2, FW3 positions 74–79, and CDRH3 positions 96–98 (Supplementary Fig. 1a) from an anti-MTX $V_{HH}$ camelid[27] antibody onto a surrogate VH domain with high sequence homology, M2J1[30]. CDRH2 position 53 and FW3 position 58 were maintained with the surrogate M2J1 sequence as structural data from $V_{HH}$ + MTX complex indicated that these positions did not appear to be involved in MTX binding[27]. The resulting scaffold, M2J1-MTX, was confirmed to retain binding and specificity to MTX (Fig. 1a), and later determined to be compatible with light-chain germlines $V_{\kappa}1$–39, $V_{\kappa}3$–20, and $V_{\lambda}1$–47. CDR1, CDR2, and CDR3 of these light chains were diversified in a manner analogous to that used to diversify the heavy-chain CDRs in the existing library[30]. The resulting scFv library was cloned into a phagemid producing a library with $1.5 \times 10^{11}$ transformants, which was used to generate a conditional scFv phage library.

**Antibody discovery**. Four rounds of phage panning were performed using a solid-phase selection by capturing biotinylated human CD33 or human EGFR onto streptavidin (Thermo Fisher Scientific) or NeutrAvidin (Thermo Fisher Scientific)-coated 96-well microtiter plates. Selective pressure was achieved using decreasing phage input and antigen concentrations with each successive round of panning. A final fifth round of panning incorporated the use of unbiotinylated target antigen as a soluble competitor to enrich for clones with high affinity and slower off-rates. After the terminal round of panning, a total of 1,536 clones for each campaign were screened for binding specificity using an automated phage ELISA. Biotinylated target antigen, MTX, and unrelated antigen (mouse 4-1BB) were captured on streptavidin-coated ELISA plates. Each phage was tested for binding to these captured biotinylated ligands, and a blank (streptavidin-coated) well. The unrelated antigen (mouse 4-1BB) and blank allowed non-specific clones to be deselected (Supplementary Fig. 13a, b). Phage showing specific target binding were selected for inclusion in inhibition ELISAs where binding to immobilized target antigen was tested in the presence and absence of 100 nM soluble target antigen, and the presence and absence of 10 µM MTX (Supplementary Fig. 13a, c). Identified phage clones were then reformatted into scFv-Fc (scFv with human Fc fused to the C-terminus). In addition, the anti-MTX $V_{HH}$ camelid antibody was also reformatted in the same manner.

**Kinetics and affinity analysis**. All kinetics and affinity parameters were determined by surface plasmon resonance. For binding of MTX to scFv-Fc and $V_{HH}$-Fc fusion proteins, the biotinylated fusion protein was captured on a streptavidin-coated sensor chip, and MTX flowed as the analyte. All kinetics and affinity analysis was performed at 37 °C. Detailed methods are provided in Supplementary Information.

**Inhibition of CD33/scFv and EGFR/scFV interactions by MTX**. Inhibition of the CD33/scFv and EGFR/scFv interactions by MTX was performed by surface plasmon resonance. Detailed methods are provided in Supplementary Information.

**Protein crystallization, data collection, and structure determination**. Conditional scFv/MTX complex was made by mixing purified scFv with methotrexate disodium salt dissolved in 1× PBS at a 1:10 molar ratio and incubating on ice for 1 h before further purification by size-exclusion chromatography over a Superdex 75 column (GE Healthcare Life Sciences) equilibrated in 1× PBS. The complexes were then concentrated for crystallization experiments. Purified apo conditional scFv or conditional scFv/MTX complex was screened for crystallization hits in sitting drop 96-well format using a Mosquito liquid handling robot (TTP Labtech). Hits were translated to hanging-drop vapor diffusion. Apo conditional scFv was crystallized in 0.1 M Tris pH 8.5, 2% Tacsimate pH 8, 16%w/v PEG 3350. Conditional scFv/MTX complex was crystallized in 0.15 M cesium chloride, 15% w/v PEG 3350. Crystals that were suitable for diffraction experiments were harvested, cryo-protected, flash cooled, and stored in liquid nitrogen for transport to the beamline. Diffraction images were collected at the Advanced Light Source beamline 5.0.2 on a Pilatus detector (Dectris) and were indexed, integrated, and scaled using XDS[47]. All diffraction experiments were carried out 100 K. Data were collected at a wavelength of 1 Å. Phases were determined by molecular replacement with Phaser[48]. Structure refinement was carried out using PHENIX[49] and structure validation performed using MolProbity[50]. Model inspection and manual rebuilding were performed using COOT[51]. Figures were generated using PyMOL. Final data collection and refinement statistics are listed in Supplementary Table 2.

**Conditional CAR construction**. Using commercial gene synthesis (Genscript), the conditional scFv nucleotide sequences were inserted prior to the CD8α hinge and transmembrane domains (aa 135–203; accession number: P01732 [https://www.ncbi.nlm.nih.gov/gene?Db=gene&Cmd=DetailsSearch&Term=925]), which are followed by 4-1BB (aa 214–255; accession number: Q07011 [https://www.ncbi.nlm.nih.gov/gene?Db=gene&Cmd=DetailsSearch&Term=3604]) and CD3ζ ITAM (aa 52–164; accession number: P20963 [https://www.ncbi.nlm.nih.gov/gene?Db=gene&Cmd=DetailsSearch&Term=919]) intracellular domains. A V5-tag (IPNPLLGLDST) was inserted between the scFv and the CD8α hinge and transmembrane domains when constructing conditional CARs targeting CD33. We additionally created a conventional CAR construct using M195-derived scFv sequence in place of the conditional scFv sequences. All of the CAR expression plasmids were constructed using a second-generation self-inactivating lentiviral vector.

**Human primary T cell isolation, lentiviral transduction, and CAR T cell production**. Human peripheral blood was obtained from anonymous healthy donors through Stanford Blood Bank and was used in accordance with Pfizer IRB/IEC policies. Peripheral blood leukocytes were isolated with Ficoll-Paque PLUS according to the manufacturer's protocol. Pan T cells were isolated via negative selection using Pan T Cell Isolation Kit, human (Miltenyi Biotec) and cryopreserved in 90% human AB serum (Gemini Bio-Products) and 10% DMSO.

Two days prior to lentiviral transduction, pan T cells were thawed and cultured in T cell transduction medium (X-VIVO 15 media, 10% Hyclone FBS, 100 IU/ml human IL-2 (Miltenyi Biotec), 20 mM HEPES, non-essential amino acids, and sodium pyruvate) with human T-activator CD3/CD28 beads (Thermo Fisher Scientific) used at a 1:2 bead:cell ratio. CAR T cells were generated using lentiviral supernatants from HEK293T cells transiently co-transfected with pLVX transgene expression vector and the viral packaging plasmids pMD2.G and psPAX2 (licensed from the laboratory of Prof. Didier Trono, EPFL) using Lipofectamine 2000 (Life Technologies). In total, $1 \times 10^6$ primary T cells (resuspended in 500 µL of T cell transduction media) were plated 2 days post-activation on a 24-well plate and mixed with 1 mL of filtered (0.45 µm) lentiviral suspension and Synperonic F108 (1 mg/ml; Sigma Aldrich). T cells were expanded in the presence of rhIL-2 (100 IU/ml) and human T-activator CD3/CD28 for 14 days prior to magnetic bead removal and cryopreservation (Supplementary Fig. 13d). During the CAR T cell production process, condCAR T cell clones were evaluated based on expansion kinetics, transduction efficiencies, degree of tonic signaling[52], and degree of T cell differentiation (Supplementary Fig. 13d–h). In order to ensure the selection of highly specific CAR clones, the conventional CAR T cells and the selected condCAR T cells were co-cultured with MV4-11mut (CD33+) or K562 (CD33−) cells to measure the cytotoxicity after 48 h (Supplementary Fig. 7).

MV4-11mut cell line development. MV4-11 cells were subcultured 24 h prior to lentiviral transduction. MV4-11 cells were first transduced with ready-to-use concentrated lentivirus encoding GFP and firefly luciferase (amsbio) and selected using the Blasticidin marker. The resulting cell line was separately transduced with lentiviral supernatants from HEK293T cells transiently co-transfected with pLVX-SFFV-hDHFRmut-IRES-Puro encoding human dihydrofolate receptor with three mutations (Q35H, F34V, F31A)[34], along with viral packaging plasmids and selected using the puromycin marker to generate the MV4-11mut cell line.

**Firefly luciferase luminescence-based cytotoxicity assays**. All in vitro culture experiments were done in RPMI-1640 media supplemented with 1% L-glutamine, and 10% FBS. No exogenous cytokines were added. All in vitro luciferase assays were performed with the One-Glo Luciferase Assay System (Promega) and 96-well Clear Bottom White Microplates (Corning), and were conducted according to the manufacturer's protocol. Target cells expressing firefly luciferase were co-cultured with T cells at specified E:T ratios in the presence of vehicle or MTX (100 µM) plus leucovorin (LV; 1 mM) in a final volume of 100 µL media for a specified duration. In experiments where MTX concentration was varied, MTX:LV molar ratio of 1:10 was maintained. After co-culture, 100 µL of One-Glo reagent was added to measure target cell viability via luminescence. Luminescence values were normalized to background cytotoxicity from NTDs.

**Flow-based T cell proliferation/activation and MSD-based cytokine analysis**. To determine target-specific proliferation, $1 \times 10^5$ CAR T cells were labeled with 1 µM CellTrace™ Violet (Life Technology) and co-incubated with either $5 \times 10^5$ target cells or $5 \times 10^4$ human T-activator CD3/CD28 beads in a final volume of 1 mL RPMI-1640 media supplemented by 10% Hyclone FBS, for 7 days at 37 °C. Cells were then recovered and labeled with anti-human CD3-BV421 (1:100; Biolegend) and anti-V5-FITC (1:200; Invitrogen). After staining, 10 µL of CountBright beads (Thermo Fisher Scientific) were added to the cells in 190 µL volume. The number of cells and beads were calculated following the manufacturer's protocol. To measure CD69 expression, cells were pelleted after a 24-h co culture (E:T = 1:2) for flow cytometry analysis, while supernatants were collected for cytokine analysis. Cells were collected and labeled with anti-human CD69-BV605 (1:100; Biolegend). All flow cytometric analysis was done on an LSRII cytometer (BD Biosciences) using FACSDiva (BD Biosciences) and analyzed with FlowJo software version 10 (Tree Star). Cytokine production was quantified by MSD U-PLEX assays (Mesoscale Discovery) according to the manufacturer's protocol.

**Flow-based cell enumeration and cytotoxicity assay**. Non-transduced T cells, conventional CAR T cells, and condCAR T cells were co-cultured with target cell lines at a defined E:T ratio either in the absence or presence (100 µM or specified concentration) of MTX. After staining for CD3 and V5-tag, 10 µL of CountBright beads (Thermo Fisher Scientific) were added to the cells in 190 µL volume. The number of cells and beads were calculated following the manufacturer's protocol.

**Long-term reversible killing assay**. To assess the reversibility of condCAR T cell antitumor activity following repeated tumor challenge, CAR T cells were added to a suspension of target cells ($1 \times 10^5$ total MV4-11mut cells, 1 mL final volume) in a 24 well plate at E:T ratio of 1-to-1. After 48 h, 50 µL of each sample was collected for flow-based cell enumeration. The remaining cells were spun down and resuspended with 1 mL of fresh media ± MTX:LV (0.1:1 mM) containing an appropriate number of target cells to maintain the E:T ratio. The resulting cell mixture was incubated for another 48 h for final time point analysis.

**Ethics statement**. All procedures were performed in accordance with regulations and established guidelines and were reviewed and approved by Pfizer's Institutional Animal Care and Use Committee.

**Mouse model and quantitative bioluminescence**. All procedures were performed on 6–8-wk-old female mice in accordance with regulations and established guidelines and were reviewed and approved by Pfizer's Institutional Animal Care and Use Committee. The mice were housed in a pathogen-free BSL2 biohazard facility with unrestricted access to water and food. The ambient temperature was restricted to 65 to 75 °F with 40–60% humidity. Mice were exposed to a 12:12 h light–dark cycle.

A total of 40 immunodeficient NSG mice (NOD.Cg-$Prkdc^{scid}$ IL-2R$\gamma^{tm1Wjl}$/SzJ) were implanted on day −8 with MV4-11mut tumor cells ($1 \times 10^6$ cells per animal in 200 µL of PBS i.v.). Tumor cells were allowed to expand until mice randomization, performed at day −1 on the basis of tumor bioluminescence. On the following day (day 0), mice were adoptively transferred (i.v.) with either $5 \times 10^6$ viable conventional CAR T cells, or $5 \times 10^6$ viable condCAR T cells, or an equal total number of NTDs (ten mice per group). At the same time, each group was divided into two subgroups, where each subgroup was implanted with an osmotic pump (Alzet; s.c.) containing 0 (PBS) or 500 mM MTX (five mice per subgroup). Leucovorin (50 mM) was also supplemented using s.c. osmotic pumps. The pumps were surgically removed the next day and replaced with fresh pumps containing the same concentrations of MTX, or PBS, and LV. On the second day (day 2), pumps were removed prior to whole-body luminescence imaging.

MV4-11mut tumor cell expansion was monitored on days −8, 0, 2, 4, 7, and 10 by bioluminescence imaging (BLI) using XenoLight D-luciferin (PerkinElmer) injected i.p. in animals (100 µL of 15 mg/mL solution per flank). Data acquisition

and analysis were performed with a Spectrum-CT apparatus (PerkinElmer) interfaced to Living Image software (Caliper).

A parallel study was used for blood collection and flow cytometry-based cell enumeration from blood. On days 2, 4, 7, 10, and 12, blood was collected by bleeding from the submandibular vein. In all, 60 μL of blood was pre-incubated with RBC lysis buffer (Biolegend) prior to incubation with TruStain FcX antibody (Biolegend). Subsequently, the cells were stained for murine CD45 (1:200; Biolegend), human CD3 (1:200; Biolegend), and V5-tag (1:200; Invitrogen). All the samples were then analyzed after adding 123count eBeads (Invitrogen) for cell enumeration.

**Statistical analysis**. Data presented as means ± SD or SEM as stated in the figure legends. Results were analyzed by Student's $t$ test (two-tailed) or by two-way ANOVA as stated in the text, and statistical significance was defined at $P < 0.01$. Pair-wise multiple comparisons were performed using multiple $t$ tests corrected for multiple comparisons with the Holm-Sidak method. $P$ values are provided in Source Data. All statistical analyses were done with Microsoft Excel 2016 and Prism software version 6.0 (GraphPad).

All reagents can be provided with the exception of the vectors used for protein expression including phage display and MV4-11mut and U87-EGFR cell lines because of existing licensing agreements that prevent this.

**Reporting summary**. Further information on research design is available in the Nature Research Reporting Summary linked to this article.

## Data availability

The atomic coordinates and structure factors have been deposited in the Protein Data Bank, www.wwpdb.org (PDB codes 6P and 63). The authors declare that all other data supporting the findings of this study are available within the paper and its Supplementary Information files. Source data are provided with this paper.

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

## Acknowledgements
We gratefully acknowledge Emma Sangalang and Colleen Brown for help in protein expression and purification. We thank German Vergara and Teresa Radcliffe and their teams for support with the animal studies. We also appreciate the help we received from Fan Yang and Pawel Dominik for carefully reading the paper and providing valuable feedback.

## Author contributions
T.V.B., J.P., B.J.S., and J.C. conceived the study. S.P., T.V.B., E.P., K.C.L., C.K., X.D., Y.S.L.M., and Z.M. performed experiments. T.V.B., K.C.L., T.O.J., R.L., B.B., R.T.A., J.P., B.J.S., and J.C. provided conceptual advice and technical support. S.P., T.V.B., E.P., K.C.L., C.K., and X.D. analyzed experiments. S.P., T.V.B., and K.C.L. wrote the paper with support from all authors.

## Competing interests
S.P. and B.B. are present Lyell Immunopharma employees. T.V.B., R.L., Z.M., Y.S.L.M., and B.J.S. are present Allogene Therapeutics employees. E.P., K.C.L., and J.C. are present Pfizer employees. C.K. is a present Asher Bio employee. X.D. is a present Dren Bio employee. R.T.A. is a present Vividion Therapeutics employee. J.P. is a present ALX Oncology employee.
