## [Peer Review File · Nature Communications]

Reviewers' Comments:

Reviewer #1:

Remarks to the Author:

This is an interesting manuscript with an ambitious but intriguing engineering strategy for CAR T cells that can be 'controlled' through the use of MTX. There are several major issue that ought to be addressed to enhance clarity and to enhance the scientific merit of the paper.

Major comments:

1. What is the role of the CDRH2 loop of the MTX-targeting mAb and why is it important to graft it into the CAR (may CDRH1 alone be sufficient, and provide sufficiently low binding affinity to MTX as desired by the authors in the Discussion)?
2. The authors ought to demonstrate that their CD33 CAR (and EGFR mAb/CAR) retains specificity for CD33 and confers selective recognition of CD33+ target cells, without (enhanced) off-target reactivity compared to the conventional CD33 CAR.
3. The authors ought to provide data at least in vitro for one additional target (with CAR T cells) to demonstrate this approach can be transferred to other applications.
4. The authors show expression of a DHFR mutant in MV4;11 tumor cells (with the intention to enable a cytotox assay and reduce background killing of the target cells in the presence of MTX?) – would it not make more sense to express the DHFR mutant variant in T cells to mitigate the toxic effect on CAR T cells? This should ideally be addressed experimentally, but at least in the Discussion.
5. The cytotoxic effect of MTX on CAR T cell viability and function (in vitro and in vivo) ought to be described in more detail to provide a perspective for the reader – to what extent does the presence of MTX diminish the viability and anti-tumor efficacy/potency upon short-term, and long-term exposure to the drug. As a reference, treatment with steroids, dasatinib or a suicide gene (like iCasp9) could be included to demonstrate where the 'sweet spot' of this technique is (transient, reversible control).
6. The authors ought to better discuss the potential clinical applicability and use of this technology. The administration of MTX is toxic, and the high serum levels required to achieve the inhibitory effect with the current CAR design (100 micromol MTX required for functional inhibition in vitro) can typically be tolerated in humans for only a short window of time (24h; then the Leucovorin rescue kicks in). This is different from alternative on/off switch techniques that have been developed by other groups to accomplish control of CAR T cell function (e.g. split CAR designs – STOP CAR / Dasatinib control) that can be engaged in the off phase for multiple days without toxicity.
7. Language and style: the manuscript contains a lot of technical detail and in the current wording may only be comprehensible by an expert audience in antibody engineering / biological chemistry. Authors are encouraged to adapt to address a wider scientific audience.

Reviewer #2:

Remarks to the Author:

CAR T cells have been approved by FDA for treating certain blood cancers have demonstrated great success in the clinic. However, CAR T cell-related cytokine release syndrome (CRS) is an ongoing challenge. Park et al aims to develop a controllable CAR T cell whose antigen recognition is modulated by methotrexate (MTX) and thus its cytotoxicity could be controlled by MTX.

The creation of an ScFv whose binding affinity is under the control of MTX is novel and desirable. MTX is already used in the clinic and if successful, this approach could be translated immediately. However, there are some main issues with this manuscript.

1. The authors demonstrated that their CAR cytotoxicity was attenuated by the addition of MTX. However, the main goal of this manuscript it to demonstrate that CRS can be controlled, which

was not analysed in vivo. Instead, tumour burden was increased after the treatment of MTX, which is not desirable.

2. In the clinic, most of the CRS is caused by IL-6. But the cytokine release assays in this manuscript investigated IL-2 and IFN-g instead.

3. This manuscript used a CAR against one cell line. It is desirable if the author could demonstrate the effect in multiple models.

4. There are a couple of duplications of figures. i.e. Figure 3f lower panels, four of the figures look identical. Sup Figure 5 panel P02-D08 and P02-E11 used identical FACS plots, but the percentage noted on the figure are different.

Reviewer comments and author responses

Reviewer #1 (Remarks to the Author): This is an interesting manuscript with an ambitious but intriguing engineering strategy for CAR T cells that can be ‘controlled’ through the use of MTX. There are several major issue that ought to be addressed to enhance clarity and to enhance the scientific merit of the paper.

Major comments:

1. What is the role of the CDRH2 loop of the MTX-targeting mAb and why is it important to graft it into the CAR (may CDRH1 alone be sufficient, and provide sufficiently low binding affinity to MTX as desired by the authors in the Discussion)?

Response: This is a great question that we failed to discuss in the originally submitted manuscript. It has now been addressed in the 1st and 4th paragraph of the Results and Discussion section of the updated manuscript as well as through the addition of Supplementary Figure 5. We hope this sufficiently addresses your question and below is a brief summary of what we added for your convenience:

Our analysis of the previously published V_{HH} crystal structure¹ indicate that CDRH2 makes contributions to the stability of the MTX binding pocket formed by the framework and CDRH1. To avoid potentially disrupting these important roles of CDRH1 and CDRH2, we decided to limit amino acid diversity to the remaining four CDRs including CDRH3 believing this would be sufficient to engineer antibodies that bind to a tumor associated antigen (TAA) while maintaining MTX binding. The results presented in the manuscript indicate that this design was sufficient to achieve the intended goal.

2. The authors ought to demonstrate that their CD33 CAR (and EGFR mAb/CAR) retains specificity for CD33 and confers selective recognition of CD33+ target cells, without (enhanced) off-target reactivity compared to the conventional CD33 CAR.

Response: We agree that specificity is a very important property of any antibody-based therapy including CAR T cells. Even though our technology will provide an additional layer of safety over conventional CAR T cells including unforeseen off-target reactivity, we employed a rigorous scFv and CAR T cell discovery process that focused on the selection of scFvs with high specificity and developability. This is now further highlighted in the Methods sections titled “Antibody Discovery” and “Human primary T cell isolation, lentiviral transduction and CAR T cell production” which reference the newly added Supplementary Figure 13. Further, the 5th paragraph of the Results and Discussion section of the updated manuscript and Supplementary Figure 7 are included to address your point about the target cell specificity of CD33 CAR T cells through use of an *in vitro* cytotoxicity assay. The newly added Supplementary Figure 10b is intended to demonstrate the specificity of the EGFR CAR T cells. We hope these additions

satisfactorily address your concerns. Further, we have added some additional context below:

After the phage panning process, we screened whole phage displaying clonal scFvs (phage clone) for binding specificity using a direct binding ELISA where the biotinylated targets were immobilized on streptavidin coated plates and phage clones were bound as the analyte (Supplementary Figure 13a). For each phage clone, we included four individual wells coated as follows: 1) the biotinylated antigen used in the phage display selection process (either CD33 or EGFR), 2) biotinylated MTX, 3) biotinylated mouse 4-1BB to assess non-specific binding and undesired binding to the tags associated with the purification and biotinylation of human CD33 and human EGFR, and 4) a blank well containing only streptavidin to further assess non-specific binding. Additional details regarding the ELISA have been added to the Methods section titled “Antibody discovery” and the results for CD33 scFvs are shown in Supplementary Figure 13b.

Phage clones determined to have the desired target specificity from the direct binding ELISA were then screened using inhibition ELISAs (Supplementary Figure 13a). In one inhibition ELISA, phage clones were tested for binding to immobilized target antigen (CD33 or EGFR) in the absence or presence of 100 nM soluble target antigen (CD33 or EGFR). Percent inhibition values were calculated to provide an affinity ranking and to ensure specificity of binding since we previously determined non-specific binders give very poor inhibition in this assay format. In the second inhibition ELISA, phage clones were screened for MTX sensitivity by testing each phage clone for target (CD33 or EGFR) binding in the presence and absence of 10 μ M MTX. Percent inhibition by MTX was calculated for each phage clone. Additional details regarding the inhibition ELISA have been added to the Methods section titled “Antibody discovery” and the results for CD33 scFvs are shown in Supplementary Figure 13c.

The most MTX-sensitive, CD33 specific scFvs were cloned into a lentiviral vector for CAR T cell production where further specificity and developability were assessed. The cytotoxicity against target-positive and target-negative cells were determined to further ensure the specificity of the scFvs. Furthermore, clones that exhibited tonic signaling² (i.e. antigen-independent signaling above that observed with the conventional CAR T cell), which can be caused by aggregation, self-interactions of scFv, or reactivity toward soluble or membrane-bound off-target entities expressed by T cells, were eliminated from further investigation³. Additional details regarding CAR T cell production and screening have been added to the Methods section titled “Human primary T cell isolation, lentiviral transduction and CAR T cell production” and the results for CD33 CAR T cells are shown in Supplementary Figure 13d-h. The data in Supplementary Figure 7 show that CD33 conditional and conventional CAR T cells exhibit similar specific lysis of CD33 expressing cells (by comparison of lysis of target-positive cells vs.

target-negative cells). Supplementary Figure 10 shows that the EGFR conditional and conventional CAR T cells also exhibit target specific cell lysis.

3. The authors ought to provide data at least *in vitro* for one additional target (with CAR T cells) to demonstrate this approach can be transferred to other applications.

Response: Thank you for the recommendation. In addition to the *in vitro* data showing MTX binding sensitivity of our EGFR scFvs in Supplementary Figure 4, we have now included Supplementary Figure 10 which shows that *in vitro* target cell lysis of EGFR CAR T cells is specific and can be suppressed using MTX. These results are highlighted in the 10th paragraph of the Results and Discussion section.

4. The authors show expression of a DHFR mutant in MV4;11 tumor cells (with the intention to enable a cytotox assay and reduce background killing of the target cells in the presence of MTX?) – would it not make more sense to express the DHFR mutant variant in T cells to mitigate the toxic effect on CAR T cells? This should ideally be addressed experimentally, but at least in the Discussion.

Response: Thank you for the suggestion. We were also hopeful that the combination of leucovorin and the DHFR mutant would further reduce the toxicity of MTX on T cells. We tested this hypothesis by comparing MTX toxicity in the presence of leucovorin to T cells and to T cells expressing the DHFR mutant. While there was a substantial rescue from MTX toxicity to T cells when leucovorin was supplemented (Supplementary Figure 8, left graph, red bars), only a small additional benefit was observed when leucovorin was added to the T cells expressing the DHFR mutant (Supplementary Figure 8, left graph, purple bars). Since this increase was marginal, we decided to proceed without implementing the DHFR mutant in T cells. We attempted to clarify these points by editing the 6th paragraph of the Results and Discussion section.

5. The cytotoxic effect of MTX on CAR T cell viability and function (*in vitro* and *in vivo*) ought to be described in more detail to provide a perspective for the reader – to what extent does the presence of MTX diminish the viability and anti-tumor efficacy/potency upon short-term, and long-term exposure to the drug. As a reference, treatment with steroids, dasatinib or a suicide gene (like iCasp9) could be included to demonstrate where the ‘sweet spot’ of this technique is (transient, reversible control).

Response: Thank you for highlighting this since we also had concerns about the effect of MTX on CAR T cell function and the effects of long-term MTX treatment on patients. We anticipate that short-term MTX treatment will be sufficient since CRS symptoms are known to resolve quickly upon therapeutic intervention⁴. To this end, we focused on the short-term effects of MTX on CAR T cell viability (*in vitro*) and function (*in vitro* and *in vivo*). We confirmed that MTX is very toxic to T cells including CAR T cells, but the addition of the FDA-approved drug leucovorin

significantly reduces this toxicity (Supplementary Fig. 8). To demonstrate that these conditions do not impact CAR T cell function, we included a conventional CAR T cell as a control in both *in vitro* cytotoxicity assays (Fig. 3f and Supplementary Fig. 9) and *in vivo* tumor models (Fig. 3g and Supplementary Fig. 11) in order to deconvolute the MTX-induced toxicity on T cells from reduced CAR T cell function resulting from our designed switch mechanism. We demonstrate that the function of a conventional CAR T cell is not reduced by the addition of MTX in the presence of leucovorin which suggests the decrease in conditional CAR T cell activity is the result of small-molecule regulation of the CAR itself and not through MTX associated toxicity. We have also added the previously excluded [MTX] = 0 μ M data points in Supplementary Figure 9 to demonstrate that *in vitro* target lytic capability of conventional and conditional CAR T cells remains intact across the full range of MTX concentrations. We attempted to highlight these points by editing the 6th paragraph of the Results and Discussion section.

While the addition of leucovorin does not completely abrogate the toxicity associated with MTX, we have shared preliminary results demonstrating the feasibility of designing structural analogues of MTX that retain affinity to the conditional antibodies, but have reduced toxicity (Supplementary Fig. 12, and newly added Supplementary Table 1). Utilizing these safer small molecules would allow extended dosing in patients for prolonged control of the engineered cells if this was determined to be necessary. These points are included in the 12th paragraph of the Results and Discussion section.

While we agree additional *in vivo* experiments could also be used to benchmark our approach to recent alternative approaches, the ongoing pandemic is limiting our ability to conduct such long-term and sophisticated experiments at the moment. We hope these other additions to the manuscript, that further describe the effect of MTX on CAR T cells *in vitro*, provide the additional clarity on this topic that we agree is needed.

6. The authors ought to better discuss the potential clinical applicability and use of this technology. The administration of MTX is toxic, and the high serum levels required to achieve the inhibitory effect with the current CAR design (100 μ M MTX required for functional inhibition *in vitro*) can typically be tolerated in humans for only a short window of time (24h; then the leucovorin rescue kicks in). This is different from alternative on/off switch techniques that have been developed by other groups to accomplish control of CAR T cell function (e.g. split CAR designs – STOP CAR / Dasatinib control) that can be engaged in the off phase for multiple days without toxicity.

Response: We agree that this is an important topic and have made several additions that can be found in the 11th, 12th, and 13th paragraphs of the Results and Discussion section. Hopefully these address your concerns and further strengthen the manuscript. We have provided below a brief summary of what we included for your convenience:

Regarding the clinical applicability of this technology, short-term (2-36 hours) continuous infusion protocols for MTX are well established and can maintain MTX serum concentrations over the range necessary to transiently inhibit the activity of the conditional CAR T cells^{5, 6}. Fortunately, intervention at the onset of CAR T cell associated CRS typically resolves within a few hours to 2 days, which aligns very well with the timing mentioned above⁴. Further, a prolonged or irreversible intervention, that many alternative “switch” technologies provide, is not desirable as the therapeutic activity of the conditional CAR T cells would be diminished during this time. To mimic this situation, we used an *in vivo* tumor model to demonstrate that dosing of MTX with leucovorin can transiently halt conditional CAR T cell activity over a 2-day duration, after which MTX can be removed to restore the anti-tumor effect. We have now included these points in the 11th paragraph of the Results and Discussion section.

Regarding alternative technologies, we feel that the transient nature of dasatinib makes it the more promising and relevant technology for us to compare to. The use of dasatinib is similar to our approach since it is small molecule that has anti-tumor activity, does not negatively impact CAR T cell manufacturing, and rapidly modulates CAR T cell activity in a reversible manner. Unfortunately, dasatinib is a promiscuous kinase inhibitor that acts by inhibiting multiple fundamental cell signaling pathways including those used by CAR T cells and is associated its own set of toxicities⁷. As a result, reducing its toxicities without simultaneously diminishing its CAR T cell suppressive effect will be challenging. However, the conditional antibody scaffold we present here only requires MTX to act as an extracellular binding partner, and not as a cytotoxic drug, which enables us to specifically reduce MTX-associated toxicities without negatively impacting its desired “switch” function. To this end, we have initiated the development of MTX structural analogues with reduced cytotoxicity that are still recognized by our antibodies (Supplementary Fig. 12, Supplementary Table 1) and inhibit engagement of target molecules.

7. Language and style: the manuscript contains a lot of technical detail and in the current wording may only be comprehensible by an expert audience in antibody engineering / biological chemistry. Authors are encouraged to adapt to address a wider scientific audience.

Response: Thank you for pointing out these shortcomings of the manuscript. We have now included additional background information and have rewritten numerous portions of the manuscript to better explain the technical details to a wider scientific audience. Particular attention was made toward explaining antibody engineering terminology which may not be commonly known outside the sphere of biologics discovery and development. Hopefully these changes address your concerns and will make the manuscript more palatable to others.

Reviewer #2 (Remarks to the Author):

CAR T cells have been approved by FDA for treating certain blood cancers have demonstrated great success in the clinic. However, CAR T cell-related cytokine release syndrome (CRS) is an on-going challenge. Park et al aims to develop a controllable CAR T cell whose antigen recognition is modulated by methotrexate (MTX) and thus its cytotoxicity could be controlled by MTX.

The creation of an ScFv whose binding affinity is under the control of MTX is novel and desirable. MTX is already used in the clinic and if successful, this approach could be translated immediately. However, there are some main issues with this manuscript.

1. The authors demonstrated that their CAR cytotoxicity was attenuated by the addition of MTX. However, the main goal of this manuscript it to demonstrate that CRS can be controlled, which was not analysed *in vivo*. Instead, tumour burden was increased after the treatment of MTX, which is not desirable.

Response: Thank you for highlighting this and we understand that our *in vivo* model may be providing an incomplete view of CRS. CRS is a cascade of immunological events initiated by the synchronous release of cytokines, such as IFN- γ and IL-2, from over activated T cells, which in turn activate neighboring myeloid cells and macrophages to release additional inflammatory cytokines, such as IL-6^{8,9}. The NSG mouse strain, which is widely used in preclinical evaluation of CAR T cells, has been shown to insufficiently mimic the role of myeloid cells in the context of CRS⁹. For this reason, myeloid-derived IL-6, though implicated in the physiopathology of CRS, was not selected as a biomarker in our study. Nevertheless, while additional *in vivo* experiments that focus entirely on CRS^{8,9} could further strengthen this manuscript, the ongoing pandemic is limiting our ability to conduct such long-term and sophisticated experiments at the moment. We hope you understand our situation.

In addition, previous clinical CAR T cell trials also suggest over activation of CAR T cells can drive CRS in an IL-6 independent manner, as some CRS patients given tocilizumab are non-responsive¹⁰. Instead, clinical data indicate that CRS severity is closely correlated with CAR T cell expansion^{8,9}. Therefore, MTX-mediated inhibition of CAR T cell expansion would be a viable strategy to suppress excessive CRS and we have used this as the read-out in our *in vivo* studies (Fig. 3g). We have added additional text to highlight this rationale in the 9th paragraph of the Results and Discussion section which hopefully addresses your concerns.

As you noted, a continued increase in tumor burden after treatment with MTX is not desirable in a patient. For the purposes of accurately characterizing our technology, we have engineered our tumor model to artificially overexpress the DHFR mutant, which in the presence of leucovorin, becomes resistant to MTX-associated toxicity. This was done to allow us to decouple MTX-associated toxicity from CAR T cell-induced tumor lysis. For this reason, during the 2 day

administration of MTX, conditional CAR T cells antitumor activity is inhibited, and the tumor burden increases (Fig. 3g and Supplementary Fig. 11). However, once the MTX dosing is completed, we demonstrate that conditional CAR T cells resume their anti-tumor activity. Further, there is a short window following MTX dosing, where tumor burden continues to increase (from day 2 to day 4). Based on our T cell quantification data (Fig. 3g), we determined this delay is the time required for conditional CAR T cells to expand to sufficient numbers to resume measurable anti-tumor activity.

In this very aggressive systemic MV4-11 tumor model, both conventional and conditional CAR T cell cohorts eventually succumb to tumor regrowth in various secondary sites (Supplementary Fig. 11). The transient expansion of CAR T cells observed (Fig. 3g) suggests the relapse may be due to the poor long-term engraftment of human CAR T cells often observed in preclinical models in the absence of homeostatic cytokine support or in combination with other therapeutic modalities (Figure 6f from Lai et al., Figure 5 from O'Hear et al., Figure 5e from Sommer et al.)¹¹⁻¹³. Multiple injections of CAR T cells or other engineering strategies to increase T cell persistence, for example through cytokine signaling^{14, 15} or by reducing T cell exhaustion^{16, 17}, can potentially overcome the eventual tumor regrowth. However, we believe our study design was sufficient to demonstrate the mechanism of our conditional scaffold design and we did not pursue these paths.

2. In the clinic, most of the CRS is caused by IL-6. But the cytokine release assays in this manuscript investigated IL-2 and IFN- γ instead.

Response: We agree that IL-6 serves as a robust biomarker for many cases of CRS. However, since IL-6 observed in the clinic is mostly from myeloid-derived cells^{9, 18} and the detailed mechanism of CRS still remains to be elucidated¹⁹, it is difficult to accurately capture the phenomenon in an *in vitro* assay. However, excessive IFN- γ and IL-2 secretion by the activated CAR T cells leads to excessive IL-6 secretion from myeloid-derived cells⁸. Therefore, we have used T cell-derived IL-2 and IFN- γ levels as markers to demonstrate that MTX can halt CAR-dependent T cell activation that may drive CRS.

3. This manuscript used a CAR against one cell line. It is desirable if the author could demonstrate the effect in multiple models.

Response: Thank you for the recommendation. We agree that highlighting more than one model would strengthen the manuscript. In addition to CD33, we have developed conditional antibodies that demonstrate switchable behavior against EGFR that use an orthogonal tumor model. In addition to the *in vitro* binding data showing MTX-sensitivity of our anti-EGFR scFvs in Supplementary Figure 4, we have now included Supplementary Figure 10 to demonstrate specific and efficient *in vitro* target cell lysis of EGFR conditional CAR T cells which can be suppressed using MTX. These results are highlighted in the 7th paragraph of the Results and Discussion section.

4. There are a couple of duplications of figures. i.e. Figure 3f lower panels, four of the figures look identical. Sup Figure 5 panel P02-D08 and P02-E11 used identical FACS plots, but the percentage noted on the figure are different

Response: Thank you for alerting us about the duplicate figures and we apologize for this oversight. In the figure you are referring to as Supplementary Figure 5, which is now Supplementary Figure 6, the P02-E08 and P02-E11 flow plots were mistakenly taken from the same source sample (P02-E08) and we have since corrected that mistake. Figure 3f lower panels do appear to be very similar but after having reviewed the raw numbers in the plots, we can assure you the figures are distinct from one another.

1. Fanning, S.W. & Horn, J.R. An anti-hapten camelid antibody reveals a cryptic binding site with significant energetic contributions from a nonhypervariable loop. *Protein Sci* **20**, 1196-1207 (2011).
2. Long, A.H. et al. 4-1BB costimulation ameliorates T cell exhaustion induced by tonic signaling of chimeric antigen receptors. *Nat Med* **21**, 581-590 (2015).
3. Ajina, A. & Maher, J. Strategies to Address Chimeric Antigen Receptor Tonic Signaling. *Mol Cancer Ther* **17**, 1795-1815 (2018).
4. Shimabukuro-Vornhagen, A. et al. Cytokine release syndrome. *J Immunother Cancer* **6**, 56 (2018).
5. Wall, A.M. et al. Individualized methotrexate dosing in children with relapsed acute lymphoblastic leukemia. *Leukemia* **14**, 221-225 (2000).
6. Goh, T.S., Wong, K.Y., Lampkin, B., O'Leary, J. & Gnarra, D. Evaluation of 24-hour infusion of high-dose methotrexate--pharmacokinetics and toxicity. *Cancer Chemother Pharmacol* **3**, 177-180 (1979).
7. Duckett, D.R. & Cameron, M.D. Metabolism considerations for kinase inhibitors in cancer treatment. *Expert Opin Drug Metab Toxicol* **6**, 1175-1193 (2010).
8. Norelli, M. et al. Monocyte-derived IL-1 and IL-6 are differentially required for cytokine-release syndrome and neurotoxicity due to CAR T cells. *Nat Med* **24**, 739-748 (2018).
9. Giavridis, T. et al. CAR T cell-induced cytokine release syndrome is mediated by macrophages and abated by IL-1 blockade. *Nat Med* **24**, 731-738 (2018).
10. Turtle, C.J. et al. CD19 CAR-T cells of defined CD4+:CD8+ composition in adult B cell ALL patients. *J Clin Invest* **126**, 2123-2138 (2016).
11. Lai, J. et al. Adoptive cellular therapy with T cells expressing the dendritic cell growth factor Flt3L drives epitope spreading and antitumor immunity. *Nat Immunol* **21**, 914-926 (2020).
12. O'Hear, C., Heiber, J.F., Schubert, I., Fey, G. & Geiger, T.L. Anti-CD33 chimeric antigen receptor targeting of acute myeloid leukemia. *Haematologica* **100**, 336-344 (2015).
13. Sommer, C. et al. Preclinical Evaluation of Allogeneic CAR T Cells Targeting BCMA for the Treatment of Multiple Myeloma. *Mol Ther* **27**, 1126-1138 (2019).
14. Krenciute, G. et al. Transgenic Expression of IL15 Improves Antiglioma Activity of IL13Ralpha2-CAR T Cells but Results in Antigen Loss Variants. *Cancer Immunol Res* **5**, 571-581 (2017).

15. Koneru, M., Purdon, T.J., Spriggs, D., Koneru, S. & Brentjens, R.J. IL-12 secreting tumor-targeted chimeric antigen receptor T cells eradicate ovarian tumors in vivo. *Oncoimmunology* **4**, e994446 (2015).
16. Seo, H. et al. TOX and TOX2 transcription factors cooperate with NR4A transcription factors to impose CD8(+) T cell exhaustion. *Proc Natl Acad Sci U S A* **116**, 12410-12415 (2019).
17. Lynn, R.C. et al. c-Jun overexpression in CAR T cells induces exhaustion resistance. *Nature* **576**, 293-300 (2019).
18. Singh, N. et al. Monocyte lineage-derived IL-6 does not affect chimeric antigen receptor T-cell function. *Cytotherapy* **19**, 867-880 (2017).
19. Murthy, H., Iqbal, M., Chavez, J.C. & Kharfan-Dabaja, M.A. Cytokine Release Syndrome: Current Perspectives. *Immunotargets Ther* **8**, 43-52 (2019).

Reviewers' Comments:

Reviewer #1:

Remarks to the Author:

The authors have satisfactorily addressed my critique.

Reviewer #2:

Remarks to the Author:

The authors have fully addressed my questions.